# Common Genetic Variation in MC4R Does Not Affect Atherosclerotic Plaque Phenotypes and Cardiovascular Disease Outcomes

**DOI:** 10.3390/jcm10050932

**Published:** 2021-03-01

**Authors:** Lisanne L. Blauw, Raymond Noordam, Sander W. van der Laan, Stella Trompet, Sander Kooijman, Diana van Heemst, Johan Wouter Jukema, Jessica van Setten, Gert J. de Borst, Anne Tybjærg-Hansen, Gerard Pasterkamp, Jimmy F. P. Berbée, Patrick C. N. Rensen

**Affiliations:** 1Department Medicine, Division Endocrinology, Leiden University Medical Center, P.O. Box 9600, 2300 RC Leiden, The Netherlands; l.l.blauw@lumc.nl (L.L.B.); s.kooijman@lumc.nl (S.K.); endo.secretariaat@lumc.nl (J.F.P.B.); p.c.n.rensen@lumc.nl (P.C.N.R.); 2Einthoven Laboratory for Experimental Vascular Medicine, Leiden University Medical Center, P.O. Box 9600, 2300 RC Leiden, The Netherlands; 3Department Medicine, Division Gerontology and Geriatrics, Leiden University Medical Center, P.O. Box 9600, 2300 RC Leiden, The Netherlands; s.trompet@lumc.nl (S.T.); d.van_heemst@lumc.nl (D.v.H.); 4Central Diagnostics Laboratory, Division Laboratory, Pharmacy and Biomedical Genetics, University Medical Center Utrecht, Utrecht University, 3584 CX Utrecht, The Netherlands; slaan3@umcutrecht.nl (S.W.v.d.L.); gpasterk@umcutrecht.nl (G.P.); 5Department Cardiology, Leiden University Medical Center, P.O. Box 9600, 2300 RC Leiden, The Netherlands; j.w.jukema@lumc.nl; 6Surgery Specialties, University Medical Center Utrecht, Utrecht University, 3584 CX Utrecht, The Netherlands; jsetten@umcutrecht.nl; 7Department Cardiology, University Medical Center Utrecht, Utrecht University, 3584 CX Utrecht, The Netherlands; gborst2@umcutrecht.nl; 8Department Clinical Biochemistry, Rigshospitalet, Blegdamsvej 9, DK-2100 Copenhagen, Denmark; anne.tybjaerg.hansen@regionh.dk; 9The Copenhagen City Heart Study, Frederiksberg Hospital, Nordre Fasanvej 57, DK-2000 Frederiksberg, Denmark; 10The Copenhagen General Population Study and Gentofte Hospital, Herlev Ringvej 75, DK-2730 Herlev, Denmark; 11Copenhagen University Hospitals and Department of Clinical Medicine, Faculty of Health and Medical Sciences, University of Copenhagen, DK-2100 Copenhagen, Denmark

**Keywords:** MC4R, genetics, cardiovascular disease, atherosclerosis, BMI

## Abstract

We analyzed the effects of the common BMI-increasing melanocortin 4 receptor (MC4R) rs17782313-C allele with a minor allele frequency of 0.22–0.25 on (1) cardiovascular disease outcomes in two large population-based cohorts (Copenhagen City Heart Study and Copenhagen General Population Study, *n* = 106,018; and UK Biobank, *n* = 357,426) and additionally in an elderly population at risk for cardiovascular disease (*n* = 5241), and on (2) atherosclerotic plaque phenotypes in samples of patients who underwent endarterectomy (*n* = 1439). Using regression models, we additionally analyzed whether potential associations were modified by sex or explained by changes in body mass index. We confirmed the BMI-increasing effects of +0.22 kg/m^2^ per additional copy of the C allele (*p* < 0.001). However, we found no evidence for an association of common MC4R genetic variation with coronary artery disease (HR 1.03; 95% CI 0.99, 1.07), ischemic vascular disease (HR 1.00; 95% CI 0.98, 1.03), myocardial infarction (HR 1.01; 95% CI 0.94, 1.08 and 1.02; 0.98, 1.07) or stroke (HR 0.93; 95% CI 0.85, 1.01), nor with any atherosclerotic plaque phenotype. Thus, common MC4R genetic variation, despite increasing BMI, does not affect cardiovascular disease risk in the general population or in populations at risk for cardiovascular disease.

## 1. Introduction

Rare genetic mutations leading to melanocortin 4 receptor (*MC4R*) deficiency are an important monogenetic cause of obesity [1]. Around 6% of children with severe early-onset (age < 10 years) obesity have a pathogenic mutation in *MC4R* [1]. Overall, over 300 different rare deleterious genetic variants in *MC4R* have been identified in obese persons [2,3,4]. The biological mechanism underlying the association between *MC4R* and obesity, has been widely studied. Notably, decreased leptin-MC4R signaling leads to a distorted energy balance, in large due to decreased satiety [5,6]. More recently, it was shown that rare to ultra-rare nonsynonymous variants in *MC4R* (allele frequency 0.0001–2%) are, next to obesity, also associated with type 2 diabetes and coronary artery disease [7].

Although rare polymorphic variants in *MC4R* have been extensively studied, less is known about the effects of common genetic variation in *MC4R* on cardiovascular outcomes. Common variants in *MC4R* have been associated with a moderately increased risk of obesity, already in the early genome-wide association studies (GWAS) on body mass index (BMI) [8,9]. Previous Mendelian randomization studies, which use single nucleotide polymorphisms (SNPs) associated with BMI as genetic instruments for the exposure [10,11,12], suggested the association between common genetic variation in MC4R and cardiovascular disease is through higher BMI [13,14,15]. However, the specific effects of common genetic variation in *MC4R* on cardiovascular disease risk and atherosclerotic plaque phenotype through effects on BMI have not been studied so far.

In the present study, we aimed to assess the effects of common genetic variation in *MC4R* on cardiovascular disease outcomes, and we studied whether these effects were influenced by clinical parameters. Moreover, we determined if common genetic variation in *MC4R* was associated with atherosclerotic plaque characteristics in patients that underwent endarterectomy.

## 2. Materials and Methods

### 2.1. Study Overview

In the present study, we analyzed data from various cohorts to assess the effects of common genetic variation in *MC4R* on cardiovascular disease outcomes; first in large population-based cohorts (Copenhagen City Heart Study (CCHS) and Copenhagen General Population Study (CGPS), *n* = 106,018; and the UK Biobank, *n* ~ 350,000) and then in an elderly population at risk for cardiovascular disease (PROspective Study of Pravastatin in the Elderly at Risk (PROSPER), *n* = 5241). Cardiovascular disease outcomes were defined as coronary artery disease, ischemic vascular disease, myocardial infarction, and stroke. The effects of genetic variation in *MC4R* on atherosclerotic plaque phenotype were determined in a fourth study comprising patients who underwent endarterectomy of the carotid artery to treat stenosis. For all populations, we selected the rs17782313 in MC4R (minor allele frequency approximately 0.21–0.25) to assess the associations. This variant was particularly selected given its previously described associations in GWAS studies [8].

### 2.2. General Population: CCHS and CGPS

The CCHS and CGPS are two comparable prospective studies including participants from Danish descent, without overlap between the studies [16,17]. The studies were approved by institutional review boards and Danish ethical committees, and were conducted according to the Declaration of Helsinki. Written informed consent was obtained from all participants. A more detailed description of the studies can be found in the Appendix A.

For the present study, data from 106,018 participants were available for analyses. In this study, we analyzed the available cardiovascular-related clinical endpoints ischemic vascular disease (18,404 cases) and myocardial infarction (5721 cases). Ischemic vascular disease was defined as either ischemic heart disease or ischemic cerebrovascular disease. Information on the diagnoses was collected and verified through a review of all hospital admissions and diagnoses entered in the Danish National Patient Registry, all causes of death entered in the National Danish Causes of Death Registry, and medical records from hospitals and general practitioners [18]. Participants with events before study entry were excluded for each specific endpoint. DNA was available from all participants. Genotyping was conducted blinded to phenotypic data. The ABI PRISM 7900HT Sequence Detection system (Applied Biosystems Inc., Foster City, CA, USA) was used to genotype the rs17782313 polymorphism using TaqMan assays.

First, to confirm the association between genetic variation in rs17782313 and measures of obesity, we determined the per-allele effect on BMI using linear regression analysis. The associations between genetic variation in rs17782313 and incident cardiovascular disease outcomes (i.e., ischemic vascular disease and myocardial infarction) was determined using Cox proportional hazard models. The standard model was adjusted for age and sex. Subsequently, we additionally adjusted for BMI. In sub-analyses, results were stratified by sex. Data were presented as hazard ratios (per-allele effect) with corresponding 95% CI. A two-sided *p*-value < 0.05 was considered statistically significant.

### 2.3. General Population: UK Biobank

To replicate our findings of the CCHS and CGPS, we repeated the analyses in the UK Biobank cohort, a prospective general population cohort. A total of 502,628 participants between the age of 40 and 70 years were recruited from the general population of the United Kingdom (UK). All participants from the UK Biobank cohort provided written informed consent, and the study was approved by the medical ethics committee. A more detailed description of the study can be found in the Appendix A.

For the present study, we restricted the analyses to the UK Biobank participants who reported to be of European ancestry, and who were in the fully genotyped datasets (*n* ~ 350,000). We were interested in the cardiovascular-related clinical endpoints coronary artery disease (7560 cases), myocardial infarction (2481 cases), and stroke (1625 cases). Participants with a history of these disease endpoints were excluded in the respective analyses. Information on diagnoses was collected through linkage with the National Health Service (NHS) hospital admissions database. UK Biobank genotyping was conducted by Affymetrix using a bespoke BiLEVE Axium array for approximately 50,000 participants. The remaining participants were genotyped using the Affymetrix UK Biobank Axiom array. All genetic data were quality controlled centrally by UK Biobank resources. For the present study, we extracted the rs17782313 polymorphism from the whole-genome data.

Similar analyses as in the CCHS/CGPS were conducted in the UK Biobank. The standard model was adjusted for age and sex. Subsequently, we additionally adjusted for BMI as derived from bioelectrical impedance analysis. In sub-analyses, results were stratified by sex. Data were presented as hazard ratios (per-allele effect) with corresponding 95% CI. A two-sided *p*-value < 0.05 was considered statistically significant.

### 2.4. Population at Risk for Cardiovascular Disease: PROSPER

PROSPER was a prospective multicenter (Glasgow, Scotland; Cork, Ireland; Leiden, The Netherlands) randomized placebo-controlled trial to assess whether treatment with pravastatin diminishes the risk of major vascular events in elderly who were at risk of cardiovascular disease [19,20]. A whole-genome-wide screening has been performed in the sequential PHASE project [21] of PROSPER, including men and women aged 70–82 years with pre-existing vascular disease or increased risk of such disease because of smoking, hypertension, or diabetes. The study was approved by the institutional ethics review boards of the participating centers, and all participants gave written informed consent. A more detailed description of the study can be found in the Appendix A.

For the present study, we analyzed data from 5241 participants. We were interested in the cardiovascular-related clinical endpoints coronary artery disease (881 cases), myocardial infarct (648 cases), and stroke (266 cases). An Endpoints Committee was responsible for the classification of all possible study end points. The committee received all annual study electrocardiograms showing serial changes, information regarding domiciliary visits or hospitalizations associated with possible cardiovascular events, and information on all deaths (including postmortem reports and/or certification of death). Genotyping was performed with the Illumina 660 K beadchip. For the present study, we extracted the rs17782313 polymorphism from the whole-genome data.

Similar analyses as in the CCHS/CGPS were conducted in PROSPER. The standard model was adjusted for age, sex and country of recruitment. Subsequently, we additionally adjusted for BMI. In a sub-analysis, results were stratified by sex. Data were presented as hazard ratios (per-allele effect) with corresponding 95% CI. A two-sided *p*-value < 0.05 was considered statistically significant.

### 2.5. Atherosclerotic Plaque Phenotype: Athero-Express Study

The Athero-Express study is a Dutch prospective cohort study, comprising patients who were newly referred to the vascular surgery departments of the participating centers for treatment of carotid or femoral artery stenosis [22]. During the procedure, blood and plaque material were obtained and samples were used for protein and immunohistochemical analyses. Informed consent was obtained prior to surgery and the study protocol was approved by the local medical ethical committee. The Athero-Express biobank study was genotyped on two different genome-wide genotyping arrays, i.e., the Affymetrix Array 5.0 (SNP5) and Affymetrix Axiom GW CEU 1 Array (AxM). After quality-control and cleaning, genotype data were available for 1439 participants and were used for analyses. For the present study, we extracted the rs17782313 polymorphism from the whole-genome data and only included data from carotid endarterectomy patients. A more detailed description of the methods can be found in the Appendix A.

The immunohistochemical analysis protocols of plaque phenotypes (i.e., macrophages, neutrophils, mast cells, smooth muscle cells, collagen content, intraplaque hemorrhage, atheroma fat content, and calcification) have been described previously [22,23,24,25], and further details can be found in the Appendix A.

First, to confirm the association between genetic variation in rs17782313 and measures of obesity, we determined the per-allele effect on BMI, using linear regression analysis. The association between genetic variation in rs17782313 and plaque phenotypes was assessed by linear regression analyses for quantitative traits and by logistic regression for binary traits. The standard model was adjusted for age, sex, cohort, year of surgery, and two principle components (PC1 and PC2). Subsequently, the standard model was additionally adjusted for BMI. In a sub-analysis, results were stratified by sex. Data are presented as additive betas (per-allele effect) with corresponding standard error (SE). By testing for nine independent plaque characteristics, the Bonferroni-corrected alpha was set at 0.0056 (0.05/9).

## 3. Results

### 3.1. Associations of MC4R Genotype with Cardiovascular Disease Outcomes in the General Population

Characteristics of the CCHS and the CGPS populations are summarized in Table 1, stratified by rs17782313 genotype. The mean age was 58 years and participants were more frequently women than men in all genotype groups. We observed no differences in characteristics between non-carriers, heterozygotes and homozygotes of the *MC4R* rs17782313-C allele, apart from BMI with an effect size of +0.22 kg/m^2^ per additional copy of the C allele (*p* < 0.001).

Figure 1 and Appendix A show the per-allele effects of rs17782313-C allele carriage on ischemic vascular disease and myocardial infarction in the CCHS and CGPS populations. There was no evidence for an association between genetic variation in rs17782313 and cardiovascular disease outcome, i.e., the hazard ratio (95% CI) for ischemic vascular disease was 1.00 (0.98, 1.03) and for myocardial infarction 1.02 (0.98, 1.07). Results remained similar after additional adjustment for BMI, or stratification by sex.

In the UK Biobank, the BMI effect per additional copy of the rs17782313-C allele was +0.25 kg/m^2^ (*p* < 0.001). Further characteristics of the UK Biobank population are summarized in Appendix A. Findings were similar to those observed in CCHS/CGPS: the hazard ratio (95% CI) for coronary artery disease was 1.03 (0.99, 1.07), for myocardial infarction 1.01 (0.94, 1.08) and for stroke of 0.93 (0.85, 1.01) per additional copy of rs17782313-C (Figure 2 and Appendix A). Results for coronary artery disease and myocardial infarction remained the same after adjustment for BMI, or stratification by sex. Only for stroke, the hazard ratio just reached the threshold of statistical significance after adjustment for BMI in the total population (hazard ratio 0.92 (95%CI 0.84, 1.00); *p* = 0.04), after stratification by sex this effect remained significant only in women (hazard ratio 0.86 (95%CI 0.74, 1.00); *p* = 0.04).

### 3.2. Associations of MC4R Genotype with Cardiovascular Disease Outcomes in an Aged Population

The characteristics of the PROSPER population are shown in Appendix A. Participants had a mean (interquartile range) age of 75 years (72, 78) in all genotype groups. The minor allele frequency (C-allele) was 0.22. Characteristics of non-carriers, heterozygotes and homozygotes of the rs17782313-C allele were comparable. The effect on BMI was confirmed, being +0.25 kg/m^2^ per C-allele (*p* = 0.006).

In Figure 3 and Appendix A, the per-allele effects of rs17782313-C allele carriage on coronary artery disease, myocardial infarction, and stroke are shown. In PROSPER, there was no evidence for an association between genetic variation in rs17782313 and any of the cardiovascular disease outcomes, also not after adjustment for BMI or hypertension, or stratification by sex.

### 3.3. Associations of MC4R Genotype with Atherosclerotic Plaque Phenotype

Characteristics of the Athero-Express study are summarized in Appendix A, stratified by rs17782313 genotype. The minor allele frequency (C-allele) was 0.25. We found no evidence for an association between rs17782313-C allele carriage and BMI (beta −0.06 kg/m^2^; *p* = 0.695). The per-allele effects of rs17782313-C allele carriage on plaque characteristics are shown in Figure 4 and Appendix A. Overall, none of the plaque characteristics showed an evident association with rs17782313-C allele carriage. Adjustment for BMI or stratification by sex did not alter the results.

## 4. Discussion

In this study, we determined the effects of a common genetic variation in the *MC4R* gene on cardiovascular disease outcomes and atherosclerotic plaque characteristics in the general population, an elderly population at risk for cardiovascular disease, and a cohort of patients that underwent endarterectomy. The minor allele frequency (rs17782313-C-allele) was 0.25 in all cohorts, which is similar to what has been described in literature [8]. Consistently between all populations, despite an effect of common genetic variation in *MC4R* on BMI, we did not find evidence for a causal effect on cardiovascular disease risk, nor plaque phenotype.

Recently, 61 nonsynonymous rare variants (variant allele frequency 0.0001–2%) mapped to *MC4R* were identified in the UK Biobank (*n* = 452,300), of which 9 were gain-of function (GoF) variants and 47 were loss-of-function (LoF) variants [7]. In contrast to the common rs17782313 variant, these variants were all nonsynonymous. Specifically, the 9 GoF variants were strongly associated with a lower BMI with an effect size of −0.38 kg/m^2^ additional GoF variant (*p* = 2 × 10^−47^), and showed a risk reduction for coronary artery disease with an odds ratio of 0.94 (*p* = 0.02) per additionally carried allele. Interestingly, the authors showed that the BMI effect size was largely determined by the effect of the mutation on β-arrestin recruitment. β-arrestin is an important regulator in the signal transduction of G protein-coupled receptors (GPCRs), like MC4R, as it prevents GPCR coupling to G-proteins thereby preventing MC4R signaling [7]. Variants that preferentially signaled through β-arrestin showed larger protective effects on BMI and cardiometabolic complications. The effect of the common genetic *MC4R* variant studied in this paper, which is located upstream on the DNA from the *MCR4* locus, on β-arrestin recruitment is not known yet, and may be an interesting topic for future studies.

In contrast, we did not find evidence for any effect of common genetic variation in *MC4R* on cardiovascular disease risk or atherosclerotic plaque phenotype. With a minor allele frequency of 0.22–0.25, a large part of the population carries at least one copy of this allele, as compared to ≤2% for the nonsynonymous rare *MC4R* variants. Possibly, the effect of common genetic variation in *MC4R* on BMI is too small to translate into an increased cardiovascular risk. Indeed, the effect on BMI in an elderly population at risk for cardiovascular disease was only +0.25 kg/m^2^ per additional rs17782313-C allele, which might be explained by dilution of the genetic effect in older individuals. Previously, a large meta-analysis combining data from GWASs on coronary artery disease showed that the common *MC4R* variant rs663129, which is in complete linkage with the rs17782313 variant (*R*^2^ = 1), showed only a moderately increased risk with an odds ratio of 1.06 (*p* = 3.2 × 10^−8^) for coronary artery disease [26]. However, the authors reported that there were no expression quantitative trait loci (eQTL) data or ENCODE features that indicated *MC4R* as the causal gene underlying the coronary artery disease susceptibility, which is in line with the null findings in our present study. Furthermore, effect sizes in our large populations—which were not included in the previous meta-analysis—were much smaller and close to null. Nevertheless, we consider the outcome of the current study as positive, as carriers of the common BMI-increasing *MC4R* variant rs17782313-C apparently are not at increased risk of developing cardiovascular disease.

The finding that the risk for incident stroke just reached the threshold of statistical significance after adjustment for BMI in the UK Biobank population, which remained significant only in women, may imply that rs17782313-C carriage, independent of BMI, reduces the risk of stroke in women. However, this finding is rather counterintuitive given the earlier described causal relation between higher BMI and increased risk of stroke [15]. In addition, this stand-alone finding was not replicated in the PROSPER cohort, and the effect just reached the level of statistical significance (*p* = 0.04), which seems to indicate that this is a likely chance finding rather than a true effect due to multiple testing. Indeed, none of the associations examined reached the level of statistical significance after multiple-test correction (not shown).

Since obesity is a risk factor of atherosclerotic plaque instability, as characterized by a high degree of inflammation, thinning, and rupture of the lesion cap [27], we studied whether rs17782313-C carriage affects atherosclerotic plaque characteristics via BMI. In line with the lack of association between *MC4R* genotype and cardiovascular disease, we did not find any effects on plaque phenotype in the Athero-Express population. In line, the minor allele frequency of rs17782313 in this population that underwent endarterectomy of the carotid artery to treat stenosis due to a late-stage atherosclerotic plaque (i.e., 0.22) was not higher than in the general population (i.e., 0.25). In case carriers of the rs17782313-C variant would be at increased risk of atherosclerotic cardiovascular disease, an enrichment of this variant in the Athero-Express population was to be expected. It should be noted that in the Athero-Express population we did not observe an association between the rs17782313 variant and BMI, which may be explained by selection bias due to an overrepresentation of patients with obesity.

The main strength of the present study is the use of a genetic instrument to determine the causal effects of common variation in *MC4R* on cardiovascular disease risk, using data from several large cohort studies, including the CCHS/CGPS and UK Biobank. These studies provided sufficient statistical power and individual level data to draw firm conclusions on associations between genetic variation in *MC4R* and cardiovascular disease risk and potential mediation via obesity, although particularly for the outcome stroke the number was rather limited given the relatively young are of the study population. A limitation of the current study is that all study populations were from European ancestry and results may therefore not be generalizable to other populations.

In conclusion, we showed that, common genetic variation in *MC4R* does not affect cardiovascular disease risk in the general population or in populations at risk for cardiovascular disease, despite a BMI-increasing effect.

## Figures and Tables

**Figure 1 jcm-10-00932-f001:**
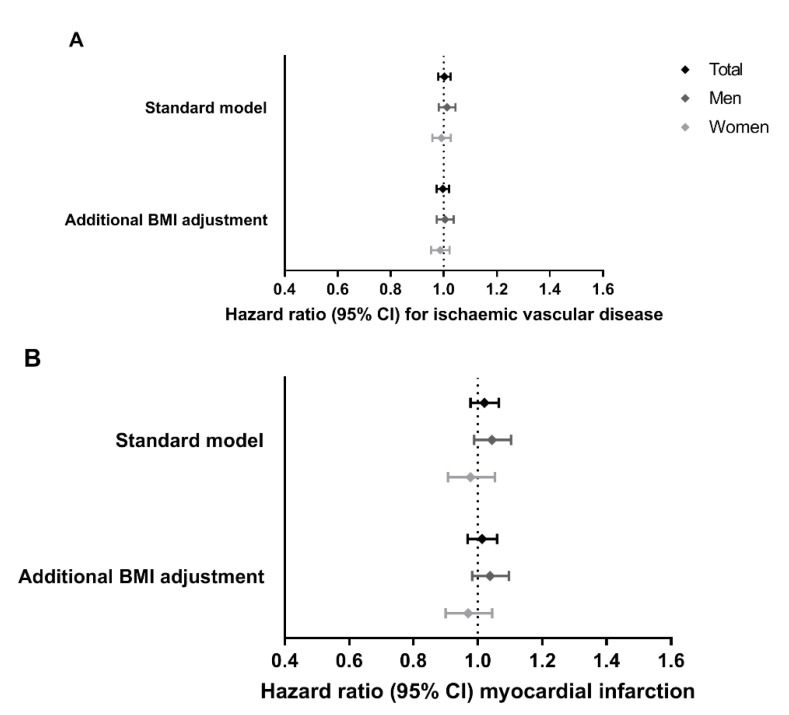
Effects of *MC4R* rs17782313-C allele carriage on incident cardiovascular disease outcomes. (**A**) ischaemic vascular disease (18,404 cases) and (**B**) myocardial infarction (5721 cases), in the Copenhagen City Heart Study and the Copenhagen General Population Study (N = 106,018 participants; 58,381 women and 47,637 men). Results from three Cox proportional hazards regression models are shown: (1) standard model, which was adjusted for age and sex (for the total population); and (2) standard model additionally adjusted for BMI. Bars represent 95% confidence intervals.

**Figure 2 jcm-10-00932-f002:**
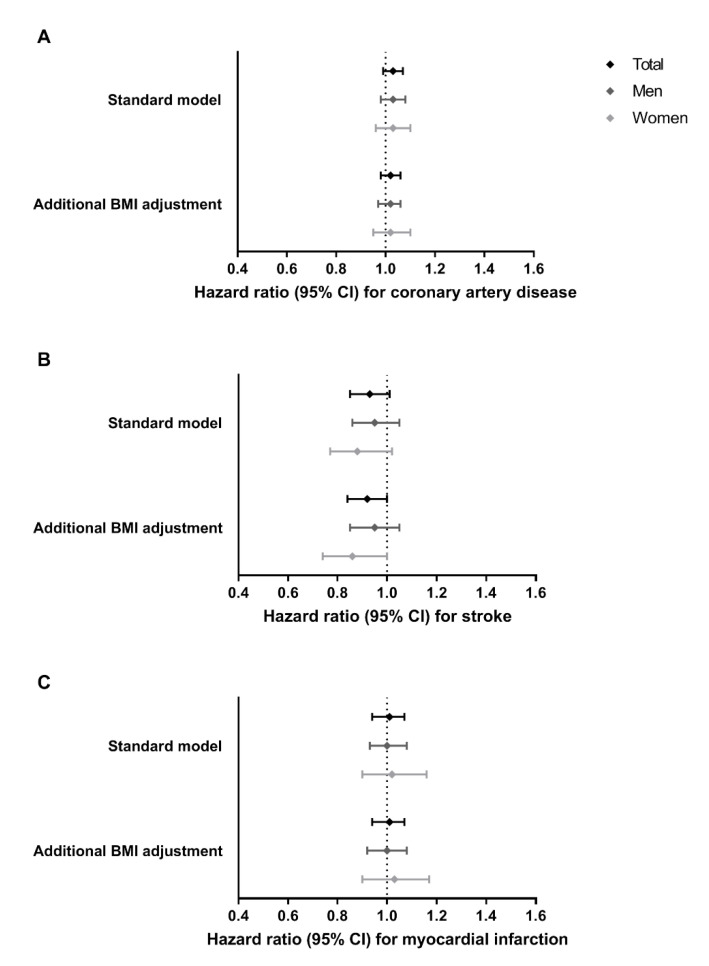
Effects of *MC4R* rs17782313-C allele carriage on incident cardiovascular disease outcomes. (**A**) coronary artery disease (7560 cases), (**B**) stroke (1625 cases), and (**C**) myocardial infarction (2481 cases), in the UK Biobank (N = 357,426 participants; 195,135women and 162,291men). Results from three Cox proportional hazards regression models are shown: (1) standard model, which was adjusted for age and sex (for the total population); and (2) standard model additionally adjusted for BMI. Bars represent 95% confidence intervals.

**Figure 3 jcm-10-00932-f003:**
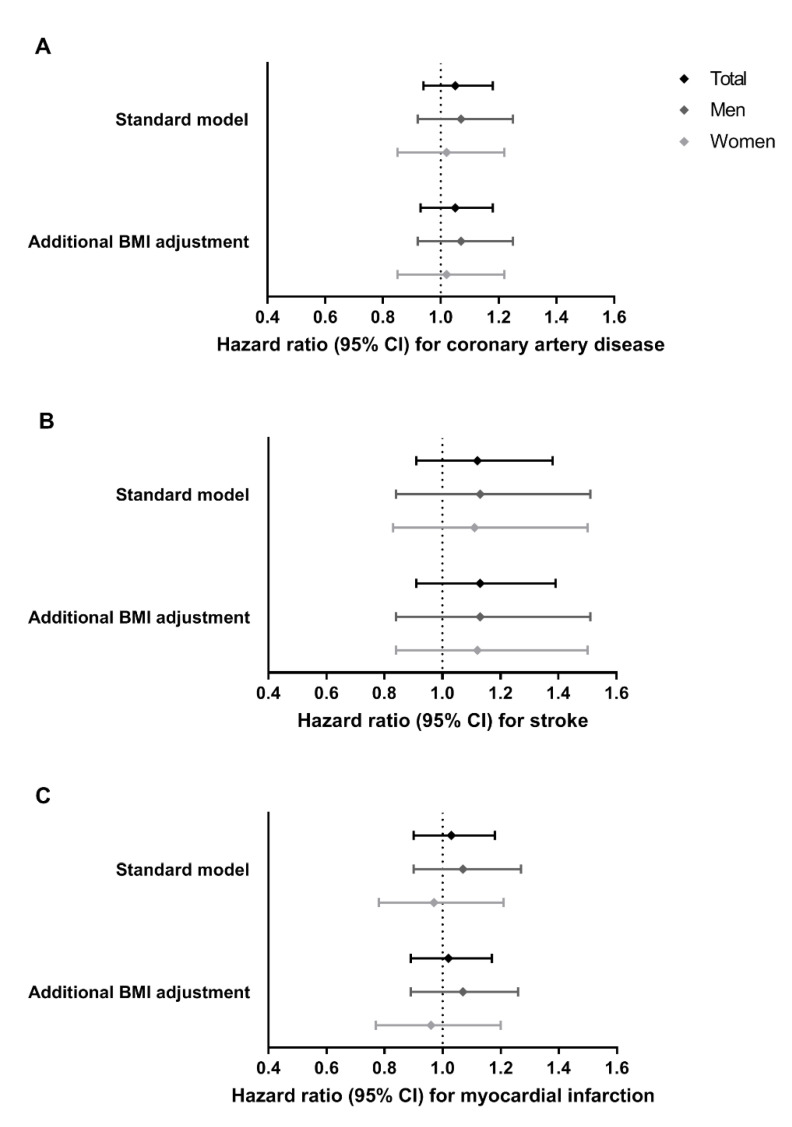
Effects of *MC4R* rs17782313-C allele carriage on incident cardiovascular disease outcomes. (**A**) coronary artery disease (881 cases), (**B**) myocardial infarction (648 cases), and (**C**) stroke (266 cases), in The PROspective Study of Pravastatin in the Elderly at Risk (PROSPER) (N = 5241 participants; 2720 women and 2521 men). Results from two Cox proportional hazards regression models are shown: (1) standard model, which was adjusted for age and sex (for the total population); and (2) standard model additionally adjusted for BMI. Bars represent 95% confidence intervals.

**Figure 4 jcm-10-00932-f004:**
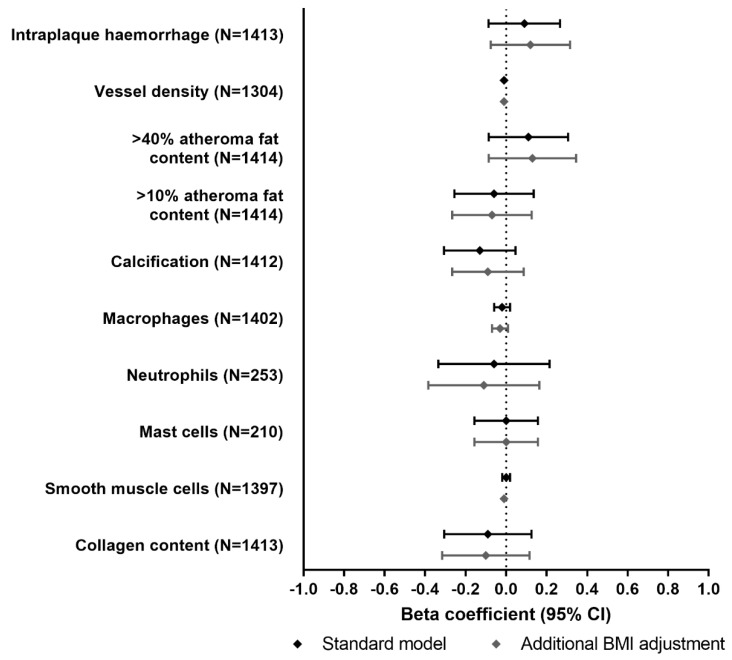
Effects of *MC4R* rs17782313-C allele carriage on atherosclerotic plaque characteristics, in the Athero-Express study (*N* = 1439; 462 women and 977 men). Results from two linear regression models are shown: (1) standard model, which was adjusted for age, sex, cohort, year of surgery and two principal components; and (2) standard model additionally adjusted for BMI. Bars represent 95% confidence intervals.

**Table 1 jcm-10-00932-t001:** Characteristics of 106,018 participants of the Copenhagen City Heart Study and the Copenhagen General Population Study, stratified by *MC4R* rs17782313-C carriage

	Non-Carriers	Heterozygotes	Homozygotes
Number of participants, N (%)	60,080 (57%)	39,426 (37%)	6512 (6%)
Women, N (%)	33,005 (55%)	21,750 (55%)	3626 (56%)
Age (years)	58 (48–67)	58 (48–67)	58 (48–68)
Body mass index (kg/m^2^)	25 (23–28)	26 (23–29)	26 (23–29)
Hypertension, N (%)	35,708 (59%)	23,552 (60%)	3894 (60%)

Values are median and interquartile range or number of subjects (N) and percentage.

## Data Availability

Data from UK Biobank can be requested by submitting a proposal to UK Biobank Recourses (www.ukbiobank.ac.uk, accessed on 1 March 2021). Data from the other studies are available upon request as part of a collaboration.

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
