# Peer review of "Common Genetic Variation in MC4R Does Not Affect Atherosclerotic Plaque Phenotypes and Cardiovascular Disease Outcomes"

_jcm, 2021, doi:10.3390/jcm10050932_

Round 1

Reviewer 1 Report

In this work, the authors study the effect of a common genetic variation (rs17782313) in MC4R gene on several cardiovascular disease outcomes (including coronary artery disease, ischemic vascular disease, myocardial infarction, and stroke) and the association with atherosclerotic plaque characteristics. To do so, the authors analyzed data from large cohorts comprising different segments of population: general population (CCHS and CGPS, n=106018; UK Biobank, n=350000) and elderly population at risk of cardiovascular disease (PROSPER, n=5241). Next, the authors studied the association of the genetic variation in MC4R with atheroma plaque characteristics in surgical pieces from patients undergoing to carotid endarterectomy (Athero-Express study, n=1439).

Obesity is a major public health concern, and it is urgent to find strategies to counteract the raising prevalence worldwide. Obesity causes are multifactorial, one of them is feeding control which determines the sensations of satiety and hunger through processes that depend on an interplay between internal signals and environmental factors. MC4R participates in these processes. Although MC4R gene mutations have been thoroughly studied and associated with obesity, the obtained results showed no association with the defined cardiovascular disease outcomes or atheroma plaque phenotype.

Despite this fact, this work is not less interesting or appealing, and the results should be taken in consideration. The study is well-designed and performed according to the hypothesis, and the results are consistent. The manuscript is well-written and presented, but the authors should carefully revise the edition (see page 4, lines 187-189).

Author Response

In this work, the authors study the effect of a common genetic variation (rs17782313) in MC4R gene on several cardiovascular disease outcomes (including coronary artery disease, ischemic vascular disease, myocardial infarction, and stroke) and the association with atherosclerotic plaque characteristics. To do so, the authors analyzed data from large cohorts comprising different segments of population: general population (CCHS and CGPS, n=106018; UK Biobank, n=350000) and elderly population at risk of cardiovascular disease (PROSPER, n=5241). Next, the authors studied the association of the genetic variation in MC4R with atheroma plaque characteristics in surgical pieces from patients undergoing to carotid endarterectomy (Athero-Express study, n=1439).

 Obesity is a major public health concern, and it is urgent to find strategies to counteract the raising prevalence worldwide. Obesity causes are multifactorial, one of them is feeding control which determines the sensations of satiety and hunger through processes that depend on an interplay between internal signals and environmental factors. MC4R participates in these processes. Although MC4R gene mutations have been thoroughly studied and associated with obesity, the obtained results showed no association with the defined cardiovascular disease outcomes or atheroma plaque phenotype.

 Despite this fact, this work is not less interesting or appealing, and the results should be taken in consideration. The study is well-designed and performed according to the hypothesis, and the results are consistent.

 COMMENT 1: The manuscript is well-written and presented, but the authors should carefully revise the edition (see page 4, lines 187-189).

REPLY: We thank the reviewer for noticing this leftover from the word JCM template. We deleted this sentence accordingly.”

Reviewer 2 Report

In this study, Blauw LL et al assessed the relationship between a common BMI-increasing MC4R allele and cardiovascular disease in different based-population studies, an elderly population study with high cardiovascular risk (PROSPER study), and in samples of atherosclerotic plaques (obtained from endarterectomy).  The main result was that, although this allele was associated with statistically significant increased BMI, no relationship with cardiovascular disease was found. This is a very interesting study, with a sound methodology.

Only minor reviews are required:

  1. Abstract: The meaning of the abbreviation MC4R should be defined. The number of patients included who underwent endarterectomy should be specified. The effect of the MC4R allele on the BMI in the present study should be mentioned as well. Finally, and although the authors have shown a summary of the main findings, some numerical data would be welcomed.
  2. In the main text, it is not clear how many patients from Athero-Express study were included.
  3. Results, lines 194-95: the comparison of the results of the allele frequency in the general population with the already describe in the literature should be moved to the discussion.  
  4. Table 1. The p-value of the comparison of the BMI among the participants according to the carriage of the minor allele should be provided.

Author Response

In this study, Blauw LL et al assessed the relationship between a common BMI-increasing MC4R allele and cardiovascular disease in different based-population studies, an elderly population study with high cardiovascular risk (PROSPER study), and in samples of atherosclerotic plaques (obtained from endarterectomy).  The main result was that, although this allele was associated with statistically significant increased BMI, no relationship with cardiovascular disease was found. This is a very interesting study, with a sound methodology.

Only minor reviews are required:

COMMENT 1: Abstract: The meaning of the abbreviation MC4R should be defined. The number of patients included who underwent endarterectomy should be specified. The effect of the MC4R allele on the BMI in the present study should be mentioned as well. Finally, and although the authors have shown a summary of the main findings, some numerical data would be welcomed.

REPLY: In line with the recommendations of the reviewer, we now defined the meaning of MC4R, added the effects of the MC4R allele on BMI, and added numerical data to the abstract.

COMMENT 2: In the main text, it is not clear how many patients from Athero-Express study were included.

REPLY: in line with the recommendation of the reviewer, we added the number of participants in the Athero-Express study to the abstract, and emphasized in the methods section that the 1,439 participants mentioned were also used for analyses.

 Changes made to the manuscript:

  • Rephrased sentence in methods section to: “After quality-control and cleaning, genotype data were available for 1,439 participants and were used for analyses.”

COMMENT 3: Results, lines 194-95: the comparison of the results of the allele frequency in the general population with the already describe in the literature should be moved to the discussion.  

REPLY: In line with the recommendation of the reviewer, we replaced the sentence to the discussion section (first paragraph).

COMMENT 4: Table 1. The p-value of the comparison of the BMI among the participants according to the carriage of the minor allele should be provided.

REPLY: The results presented in table 1 are only intended to present the current study population. As we did not test any of the study hypotheses, it is not useful to perform statistical tests. In addition, the use of p-values is much discouraged by many of the medical journals given that p-values are much dependent on the sample size. (Why not to (over)emphasize statistical significance. Olaf M Dekkers. Eur J Endocrinol. 2019 Sep;181(3):E1-E2.) As the data presented in Table 1 are derived from a large study population, many of these tests would have a p-value below 0.05 without further insights in the actual difference between the genotype groups. Together with the actual function of the Table 1 (describing the study population), we strongly believe performing any statistical tests would be inappropriate.